# Peptide Linker Affecting the Activity Retention Rate of VHH in Immunosorbents

**DOI:** 10.3390/biom10121610

**Published:** 2020-11-27

**Authors:** Da Li, Jun Ren, Fangling Ji, Qiang Peng, Hu Teng, Lingyun Jia

**Affiliations:** Liaoning Key Laboratory of Molecular Recognition and imaging, School of Bioengineering, Dalian University of Technology, No.2 Linggong Road, Dalian 116023, China; lidairis@mail.dlut.edu.cn (D.L.); renjun@dlut.edu.cn (J.R.); fanglingji@dlut.edu.cn (F.J.); truelybe@mail.dlut.edu.cn (Q.P.); tenghu@dlut.edu.cn (H.T.)

**Keywords:** peptide linker, nanobody, immunosorbent, FGE, β2-microglobulin

## Abstract

VHH-based immunosorbents are an emerging and promising tool for the removal of toxic substances from plasma. However, the small size of VHHs is a double-edged sword, bringing both benefits and drawbacks to the immunosorbent. The small size of the VHH allows a higher coupling density, while the closer distance to the resin might create steric hindrance for paratope access. The latter could be avoided by inserting a linker between the VHH and the gel attachment site. Here, we report an approach to improve the activity retention of the immobilized VHH by selecting suitable linkers between the VHH and the site-specific immobilization site on the resin. Seven peptide linkers differing in length and flexibility were fused to the VHH and contained the formylglycine generating enzyme (FGE) recognition sequence. These constructs were expressed in the cytoplasm of bacteria and purified, the VHH production yield and affinity for its cognate antigen was measured. Furthermore, the fGly conversion, the immobilization of the aldehyde-containing nanobodies, the immobilization on resin and the antigen binding activity of the VHH-based immunoadsorbents was monitored. The VHH with longer and rigid, proline-rich linkers exhibited good expression yield of approximately 160 mg/L of culture, a fGly conversion of up to 100%, and the highest activity retention rate of more than 68%. This study unveiled two suitable linkers for the preparation of VHH-based immunosorbents that will assist the development of their clinical application.

## 1. Introduction

Antibodies are increasingly being used for the preparation of immunoadsorbents due to their capacity to capture antigenic proteins with high affinity and specificity. Such immunoadsorbents can be employed for the selective removal of target proteins (e.g., pathogenic antigen) from plasma [1,2,3]. As such, several immunoadsorbents have been developed to capture uremic middle molecule toxins, such as β2-microglobulin (β2MG). Vallar et al. [4] and Ammer et al. [5] used murine anti-β2MG monoclonal antibodies (mAbs) in their immunosorbents to remove β2MG from blood and a binding capacity of 0.13 mg β2MG per mL of resin has been reported. Subsequently, single-chain antibodies (scFvs) have been introduced as immunoadsorption ligands, which increased the binding capacity to 0.41 mg/mL [6]. Our preliminary work employed nanobodies (also known as VHH) as affinity ligands to prepare a novel immunosorbent after site-specific immobilization, whereby the removal capacity of β2MG from blood was increased further to 0.75 mg per mL of resin [7]. 

The site-specific immobilization chemistry of the VHH to resin, avoiding the disruption of the VHH structure and function, is vitally important for obtaining a maximal capacity of the immunosorbent. The formylglycine generating enzyme (FGE) has emerged as a robust tool for site-specific protein modification [8,9]. FGE recognizes the amino acid sequence motif LCXPXR (referred to as “aldehyde tag”) and catalyzes the cysteine to an aldehyde-containing formylglycine (fGly) residue [10]. The aldehyde group is absent in native proteins, and it has been demonstrated as one of the most versatile handles for bioconjugation, especially suitable for the immobilization of VHH-based immunosorbent. 

Despite the interest in FGE for protein engineering, its practical use in batch-mode catalysis is inefficient. We hypothesised that the catalytic efficiency of FGE was dependent on the degree of the aldehyde tag exposure as indicated by the variable modification efficiencies when inserting the aldehyde tags at various locations within the protein sequence [11]. It seems that the aldehyde tag needs to be maximally exposed to contact the FGE. Thus, we preferred testing various types of linker (or spacers) rather than just elevating enzyme levels or reaction time to raise catalytic efficiency.

Peptide linkers do not only naturally occur in multidomain proteins, where they serve as spacers, but are also used as a powerful approach for the construction of fusion proteins or multidomain protein assemblies. In nature, the linkers connecting different modules of protein kinase systems have been found to play an important role in maintaining cooperative protein–protein interactions [12,13]. However, the majority of peptide linkers have been used in protein engineering or drug design to construct fusion proteins to match their particular biological purpose [14,15]. Due to the complexity of protein structure and function and a lack of universal rules to guide linker selection, information is lacking to identify the most appropriate peptide linker for VHH immobilization. 

The selection of the linker sequence is particularly important for the construction of functional proteins equipped with an aldehyde tag, as the flexibility and length of the linker are crucial to maintain the full functionality of the domains. George et al. [16] extracted 638 multidomain protein chains and analyzed their length, the linkers were categorized into several groups: small, medium and large linkers having average lengths of 4.5 ± 0.7, 9.1 ± 2.4, and 21.0 ± 7.6, respectively. Thus, the selection of the length of the peptide linker should be of sufficient length to avoid steric hindrance, which might decrease the activity of the functional domains. Conversely, the length of linker should not be too long to avoid intramolecular or intermolecular entanglement. 

Apart from the length, also the flexibility or rigidity of linkers should be taken into account. Flexible linkers permit a necessary degree of freedom between the joined functional domains. Flexible linkers are generally composed of small, non-polar or polar residues including Gly and Ser, with a common (Gly_4_Ser)_n_ motif. Such linkers are unstructured and, according to a previous study, provide limited domain separation [17]. By contrast, when the spatial separation of domains is critical for their bioactivity, rigid linkers are preferred to separate the functional domains and keep a fixed distance between them. Proline-rich linkers are more rigid and possess a more extended conformation [18], and occur naturally in various antibody sub-isotypes.

In this study, we choose the 5-amino-acid peptide GGGGS and its three repeats peptide GGGGSGGGGSGGGGS as flexible linkers, the natural camelid IgG_2_ hinges AHHSEDP and EPKTPKPQPQPQPQPQPNPTTE and human IgA hinge STPPTPSPSTPP as rigid linkers. In addition, a hybrid linker composed of (G_4_S)_3_ and AHHSEDP was designed. Different linker compositions can alter their effective length and rigidity. In this study we evaluated the effects of inserting the aldehyde tag C-terminally of these six linkers joined to nanobodies, and tested the expression yield of these constructs, the effect on the affinity of the nanobody moiety, the fGly conversion efficiency and the immobilization of aldehyde-nanobodies. Finally, and of utmost importance, the effect on the retention rate of antigen binding activity was assessed. The above factors jointly determine the adsorbing capacity of VHH immunosorbents. In summary, this empiric comparison of linkers enables us to identify the ideal linkers for the preparation of best performing VHH-based immunosorbents.

## 2. Materials and Methods

### 2.1. Plasmid Construction

The expression of VHH fusion proteins from the pET23a vector in Rosetta-gamiB (DE3) pLysS strain was as described before [19].

The amino acid sequence of the VHHs was: CNb1–linker–HHHHHHGGGGSLCTPSR where LCTPSR is the recognition sequence of FGE.

The amino acid sequences of the designed linkers were: no linker (A); short hinge linker (B), AHHSEDP (7 aa) derived from the camelid IgG_2c_ hinge; long hinge linker (C), EPKTPKPQPQPQPQPQPNPTTE (22 aa) derived from the camelid IgG_2a_ hinge; G_4_S linker (D), GGGGS (5 aa); (G_4_S)_3_ linker (E), GGGGSGGGGSGGGGS (15 aa); long linker (F), AHHSEDPGGGGSGGGGSGGGGS (22 aa); and the IgA linker (G), STPPTPSPSTPP (12 aa) derived from the human IgA hinge.

### 2.2. Expression of the Anti-β2MG VHHs

Rosetta-gamiB (DE3) pLysS cells harboring the constructed vector were inoculated in 1 L of TB medium supplemented with the appropriate antibiotics (100 μg/mL ampicillin, 12.5 μg/mL tetracycline, 34 μg/mL chloramphenicol and 15 μg/mL kanamycin), and shaken at 200 rpm at 37 °C. The expression of the recombinant protein was induced with isopropyl β-d-1-thiogalactopyranoside (IPTG) with a final concentration of 0.25 mM when the OD_600 nm_ of the culture reached 3.0, and incubation at 170 rpm and 18 °C was continued for 16 h. The bacterial cells were harvested, washed, and resuspended with the equilibration buffer (10 mM PBS buffer, pH 7.4) at 4 °C.

### 2.3. Purification of the Recombinant Anti-β2MG VHHs

The cells were resuspended in lysis buffer (10 mM imidazole, 20 mM NaH_2_PO_4_/Na_2_HPO_4_, 0.5 M NaCl, 0.2 mM PMSF (phenyl methyl sulphonyl fluoride), pH 7.4) at 1:8 (*w*/*v*), and lysed by a high-pressure homogenizer (APV Deutschland GmbH, Lübeck, Germany). Cell debris was removed by centrifugation at 15,000× *g* for 20 min. The supernatant was loaded onto a 5 mL HisTrap column (GE Life Sciences, Marlborough, MA, USA) using an AKTA FPLC system (GE Life Sciences, USA). The column was washed with a linear gradient of imidazole (up to 500 mM), and all eluted fractions containing target protein were collected for further purification. Final purification was performed by size-exclusion chromatography (SEC) using a Superdex 75 (10/300) column (GE Life Sciences, USA) in a buffer (10mM PBS, 0.2mM PMSF, pH 7.4). The protein concentration was determined using the Bradford protein reagent using bovine serum albumin as standard (Solarbio, Beijing, China). The processes of expression and purification of VHHs were followed and analyzed using denaturing sodium dodecyl sulfate polyacrylamide gel electrophoresis (SDS-PAGE). Protein purity was quantitatively analyzed as described by Li et al. [19] using Image Lab^TM^ software (v3.0, Bio-Rad, Hercules, CA, USA).

### 2.4. De Novo Structure Prediction for Linkers

The structure prediction of the 7 different linkers was performed by PEP-FOLD3 [20] (available online: https://bioserv.rpbs.univ-paris-diderot.fr/services/PEP-FOLD3/).

### 2.5. Anti-β2MG VHHs Affinity Measurement

The affinity of the VHHs was determined by Biacore T200 (GE Life Sciences, USA) using Control software version 2.0.2 and Evaluation software version 3.1 for interaction analysis.

The antigen β2MG was immobilized on a CM5 chip at pH 5.0 using the amine coupling kit (GE Life Sciences, USA). VHHs were injected at different concentrations (200 nM–0.78 nM in 2-fold serial dilutions) into a running buffer (HBS-EP+ (10 mM HEPES, 150 mM NaCl, 3 mM EDTA, 0.05% P20), pH 7.4). The association phase was monitored for 200 s, and the dissociation phase was monitored for 300 s. The chip surface was regenerated after each cycle by injecting a 10 mM glycine-HCl buffer, pH 1.5 (30 µL/min, 45 s). The association rate constant *k_a_* and dissociation rate constant *k_d_* were calculated and analyzed using the monovalent analyte model, and the equilibrium dissociation constant (*K_D_*) was calculated (*K_D_* = *k_d_*/*k_a_*).

### 2.6. Generation of the Aldehyde Group

FGE was prepared in house and coupled to resin as described previously [21]. The resin with immobilized FGE was used to simplify the subsequent protein separation process. 

VHHs were diluted into the reaction buffer (50 mM triethanolamine, pH 9.0, 150 mM NaCl, 1.6 mM Tris (2-carboxyethyl) phosphine (TCEP)) to a final concentration of 600 μM. Then the reaction was initiated by adding 1 mL resin with immobilized FGE (39 mg/mL resin) and rotated in an end-over-end mixer for 4 h at room temperature. Previous studies showed that Cu (II) played a crucial role in the catalytic activity of FGE [22,23,24], so 0.25 mM CuSO_4_ was added to the reaction to enhance the activity of FGE.

### 2.7. Preparation of VHH Based Immunosorbents

As shown in Scheme 1, Sepharose CL-6B (40 mL, GE Healthcare, Pittsburgh, USAcity, country) was washed with water to remove storage preservatives, then drained and placed in a flask to which, 40 mL NaOH (2 M), 120 mL DMSO and 20 mL epichlorohydrin were added sequentially. The mixture was incubated at 39 °C for 2 h. The gel was washed with acetone, followed by further washing with excess of water. The activated gel was then incubated with 10% (*v*/*v*) 3,3′-Diaminodipropylamine (DADPA) at 40 °C for 4 h to generate the amino group. Thereafter, 80 mL 6% ethanolamine (pH 8.6) was added to the gel to block the remaining epoxy groups.

The VHH-CHO (3 mL, 600 μM) was added to 1 mg DADPA-modified gel, which was washed with reaction buffer (pH 9.0). The coupling process was performed at 37 °C for 12 h, followed by washing with 3 times the column volume of the reaction buffer. A further wash step containing 1 M NaCl was included to remove nonspecifically adsorbed protein from the gel. The content of protein that was coupled to the gel or that was contained in the eluent could be estimated from the difference in protein contents present in the reaction medium before and after coupling. The Schiff base bond was reduced with sodium cyanoborohydride solution to prevent leakage of the VHH from the gel. Finally, the VHH-based immunosorbent was stored at 4 °C in PBS containing 0.02% sodium azide.

### 2.8. Adsorption Performance of the VHH-Based Immunosorbents

To optimize the adsorption of β2MG, different initial concentrations of β2MG (0, 0.1, 0.2, 0.3, 0.4, 0.5, 1, 2.5 mg/L) in PBS were tested. The gel was washed with PBS and then mixed with 400 μL β2MG and incubated at room temperature under continuous rolling for 30 min. The Bradford assay was used to determine the concentration of β2MG before and after adsorption.

The β2MG adsorption capacity and $K_{d}$ of the VHH-based immunosorbent were subsequently calculated, using the Langmuir adsorption isotherm (Equation (1)) and its rearranged equation (Equation (2)).
(1)$$q=\frac{q_{m}C}{K_{d}+C}$$
(2)$$C=q_{m}\frac{C}{q}-K_{d}$$
where C (mg/mL) is the equilibrium concentration of β2MG; q (mg/mL) is the adsorption capacity when the adsorption reaches equilibrium; $q_{m}$ (mg/mL) represents the maximum adsorption capacity of the gel and $K_{d}$ represents the Langmuir adsorption constant.

## 3. Results and Discussions

### 3.1. Expression and Purification of Anti-β2MG VHH with Different Linkers

The end-to-end fusion technique is a common method to generate multifunctional protein assemblies. The linker sequence, length and flexibility are important parameters determining indisputably the functional outcome of such constructs. However, the effect of different linkers on the protein expression remained understudied. 

For the (G_4_S)_n_ linkers, high glycine content had been shown to be resistant to proteolysis during expression, whereas serine improved the solubility of the linker region in aqueous solutions [25,26]. Moreover, nucleotide sequences encoding (G_4_S)_n_ have to be designed with care. The codon of glycine is GG*N* (*N* = A, T, C, or G). Consequently, a stretch of four glysines will be encoded by a dG/dC rich sequence, so that the mRNA described from this gene might form stable mRNA secondary structures that will reduce the translation [27]. Another concern is the effect of preferred codon utilization bias that may decrease the translation efficiency [28]. In order to overcome the factors mentioned above, the stretch of (G_4_S)_n_ in CNb1-D, CNb1-E, and CNb1-F were optimized based on the preferred codon usage rule of *E. coli*.

As shown in Appendix A, a successful expression and purification of recombinant VHHs with different linkers was achieved. It was demonstrated that all our VHH constructs reached a purity above 95% after SEC. The final yields of soluble VHHs and the molecular weights of the VHHs obtained by mass spectrometry are summarized in Table 1. The yields of these soluble VHHs were much higher than the average yield of other reported VHHs [29,30]. Higher yields facilitate the preparation of immunosorbents, which require a large amount of VHH.

The expression results of these VHH with linkers were wildly divergent, which confirmed the case-dependency of recombinant protein production [15,31,32]. Interestingly, the yield of CNb1-E was inferior to CNb1-D, while the yield of CNb1-F was superior to CNb1-D (Table 1).

### 3.2. De Novo Structure Prediction for Linkers

The structure of our linkers was predicted by the algorithms of PEP-FOLD3 [20]. The models were sorted with the coarse -gained protein force filed optimized potential for efficient structure prediction (sOPEP). From the repertoire of the predict peptide models, we selected the ones with the lowest energy to calculate the distance between N-terminal end of the linker and C-terminal end of the FGE-tag (with PyMol). The prediction results were in line with our expectations (Figure 1 and Appendix A). Linkers C and G were with a high proline content were spacing the end of the VHH and the FGE tag at a longer distance than the E tag of intermediate amino acid length. This fits the idea of proline-rich sequences possessing certain rigidity, although two adjacent prolines may be necessary to significantly restrict conformational flexibility [33]. Appendix A exhibits the 20 most likely structures. The stretched conformers of linker C matched closely each other confirming its rigidity. Thus, the rigid linkers are expected to separate the VHH and the FGE recognition sequence so that the FGE-tag will be exposed and maximally accessible for the enzyme. Conversely, linkers E and F were long and expected to be highly flexible due to their high glycine content in the (G_4_S) part. For such poorly structured linkers with a high degree of rotational freedom, it is difficult to predict reliably the impact on the spacing between the VHH and the FGE-tag. The predicted distances were variable E ranging from 7.2 Å to 24.9 Å and F ranging from 12.1 Å to 25.1 Å, also indicated the flexibility of these linkers. Linkers B and D were short linkers, resulting in a close distance between the VHH and FGE-tag, whereby the D linker is supposedly more flexible than the B linker construct, which was evidenced with a longer predicted distance.

### 3.3. Anti-β2MG VHH Affinity Measurements

The main function of a linker is to provide a spatial distance between the domains while maintaining their distinct fold and functionality. To analyze the binding activity of the VHH to its cognate antigen, the affinity of VHHs with different linkers was measured by surface plasmon resonance (SPR). As shown in Figure 2 and Appendix A, the K_D_ values of all seven VHH constructs were of the same order of magnitude (10 nM), suggesting that the different linkers at the C-terminal end of the VHH did not affect the binding affinity of the VHH to its antigen. Therefore, the choice of linker is neutral for the VHH affinity to its target. In contrast, as indicated in previous studies [34,35] it has been shown that sequences upstream of the VHH (at the N-terminus of the VHH) might affect the antigen-binding capacity since the paratope of the VHH is located at the same side of the N-terminal end in the folded nanobody.

### 3.4. fGly Conversion

The reaction product (aldehyde) of FGE on the aldehyde tag fused to VHHs, can be identified and quantified using high-performance liquid chromatograph-high resolution mass spectrometry (HPLC-HRMS) analysis [36,37]. As shown in Figure 3, the fGly conversion of the FGE-tag was related to the structure of the linkers under the determined reaction conditions. The long and rigid linkers (C and G) were conducive to expose the FGE-tag adequately, whereby the reaction efficiency was maintained at 100% (equal to the efficiency of the CNb1-A, which was taken as reference). In contrast, the FGE-tag downstream of the long and flexible linkers (E and F, which contained GGGGSGGGGSGGGGS in their sequence) was apparently less accessible for the FGE reaction. This reinforces the idea that unstructured, flexible linkers provided limited domain separation, which compromised the accessibility and reactivity of the FGE-tag [17]. In addition, this linker is prone to intramolecular or intermolecular entanglement [38] and wrapping the FGE-tag at an inside location. Appendix A showed that E and F coil and bend the oligopeptide so that the two ends come closer together. Thus, the long and flexible linkers were not conducive to the catalysis of the FGE-tag by FGE and the reaction efficiencies was reduced to 87.5% and 92.4%, respectively. With short linkers B and D, the surrounding domains are less separated than with long linkers, which should provoke hindrance to the accessibility of the FGE-tag, leading to a reduced reactivity. However, D was more flexible than B and it appeared to be a favorable situation that facilitated the reactivity of the FGE-tag. The fGly conversions were consistent with the structural prediction, the shorter and more flexible linker D happened to extend a longer distance and provided the accessibility of the FGE-tag. 

### 3.5. Immobilization of VHH-Aldehyde

Considering the low expression yield, we decided to stop further use of CNb1-E. Moreover, CNb1-F was discontinued, as its fGly conversion was lower and in two attempts to immobilize this nanobody we obtained disappointing low yields (0.90 ± 0.17 mg CNb1-F/mL gel).

During the immobilization process, the VHH-aldehyde had been coupled onto the amino-activated agarose beads, which was previously synthesized at a ligand density of 66 μmol/mL. The density of protein coupling onto the gel was calculated by comparing the VHH concentration before and after immobilization. The results showed a varying immobilization efficiency of VHHs with different linkers.

As shown in Figure 4, there was no significant difference between CNb1-A and CNb1-B in their immobilization performance of about 14–15 mg per mL gel. This was significantly different from the three other samples, which yielded 6.3, 7.0 and 5.3 mg/mL for CNb1-C, CNb1-D and CNb1-G, respectively. A higher coupling density in the gel was observed for nanobodies with the shorter linkers. It seems that with shorter linkers a more spherical and integrated protein is obtained, which increased the collision probability of the two particles, VHHs and the gel, resulting in higher coupling efficiency and density. 

It should be noted that the VHH density could be controlled easily by changing the reaction time and material input [39]. Here we choose a moderate material input to avoid wasting protein, the immobilization performance under the present condition was fairly good compared with previous work that generated a VHH density of 1.2 mg/mL gel [7]. The performance we reported here was expected to meet the requirements for subsequent applications, i.e., immunosorbents.

### 3.6. Adsorption Performance of the VHH-Based Immunosorbents

To investigate the active VHH coupled onto the gel, we measured the amount of β2MG bound per mL of immunosorbents to represent the retention capacity of the VHH on the gel.

As shown in Appendix A, the adsorption of β2MG on the immunosorbents fitted very well with the Langmuir adsorption isotherm. All the correlation coefficients (R^2^) of the rearranged Langmuir adsorption isotherm model for β2MG were above 0.99, indicating that the adsorption of β2MG on the VHH-based immunosorbent was consistent with the Langmuir adsorption isotherm model and the adsorption process followed that of a monolayer process. Meanwhile, the maximal adsorption capacity of the gel and the *Kd* value of the immunosorbents were shown in Table 2 as calculated according to the fitting equations.

The immunosorbents demonstrated a high capacity and activity towards β2MG. This binding capacity of these immunosorbents was remarkable, and a much higher binding capacity of β2MG could probably be achieved by changing the coupling conditions. Anyway, even without much optimization of the coupling conditions, a massive improvement in β2MG binding capacity to 2.48–5.68 mg/g resin was noticed, in comparison to previous studies that reported 0.13 to 0.75 mg β2MG per mL resin [4,5,6,7]. We think this β2MG adsorption improvement is due to (i) the different format of the antigen binder (VHH versus scFv and mAb), (ii) the smaller size, robustness of the VHH and (iii) the different coupling chemistry allowing a directional immobilization of the probe at a higher density. 

Theoretically, the adsorption of β2MG to VHHs should be equimolar, but in practice, a loss of activity of the VHH during the coupling process might be expected. The inactivated VHH fails to bind antigen, resulting in a reduced column efficiency and an increased cost of the adsorbent. The retained antigen binding activity of the immobilized antibody was crucial for the immunosorbent, which determined the efficiency of the immunosorbent.
(3)$$\mathrm{Activity}\;\mathrm{retention}\left(\%\right)=\frac{n_{EX}}{n_{TH}}$$

where $n_{EX}$ (mol) is the test antigen adsorption data, $n_{TH}$ (mol) is the theoretical antigen adsorption capacity. 

The activity retention percentage (calculated by Equation (3)) of VHHs without linker (CNb1-A) and those with 5–7-amino-acid linkers (CNb1-B and CNb1-D) was close to 50%. In contrast, CNb1-C and CNb1-G with longer and rigid linkers, had activity retention percentages increased by more than 1.36 times. It indicated that a long and rigid linker could play the role of a spacer arm for the immunosorbent; the spacer arm offered the VHH-based immunosorbent an extended structure. This result possibly reduces steric hindrance effect and increase the activity retention percentages. 

## 4. Conclusions

In summary, while the expression yields of VHHs with long and rigid linkers C and G (around 150 mg/L culture) were reduced by 25–30% compared to VHHs without linker (CNb1-A), these constructs exhibited an fGly conversion of up to 100%, and an activity retention percentage of more than 68%. Although the immobilization efficiency of CNb1-C and CNb1-G on the activated resin was lower compared to constructs CNb1-A and CNb1-B, which resulted in inferior antigen adsorption, we think this could be improved relatively easily by changing the coupling reaction conditions. The characteristic of higher activity retention percentage made the long and rigid linkers as the ideal linkers for VHHs with an FGE-tag, especially for preparing VHH-based immunosorbents for therapeutic applications.

## Figures and Tables

**Scheme 1 biomolecules-10-01610-sch001:**
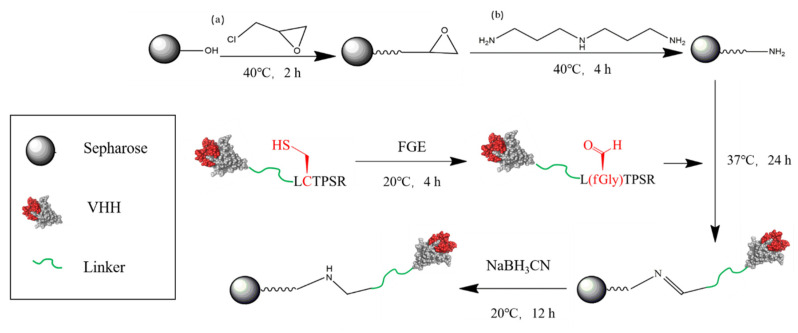
Scheme of the preparation of the VHH based immuno-adsorbent: (**a**) epichlorohydrin and (**b**) 3,3′-diaminodipropylamine were used to prepare the amino-activated matrix from CL-6B agarose beads. Aldehyde-modified VHHs were then coupled to the matrix via the reductive amination. The red region represents the complementarity-determining region of the VHH.

**Figure 1 biomolecules-10-01610-f001:**
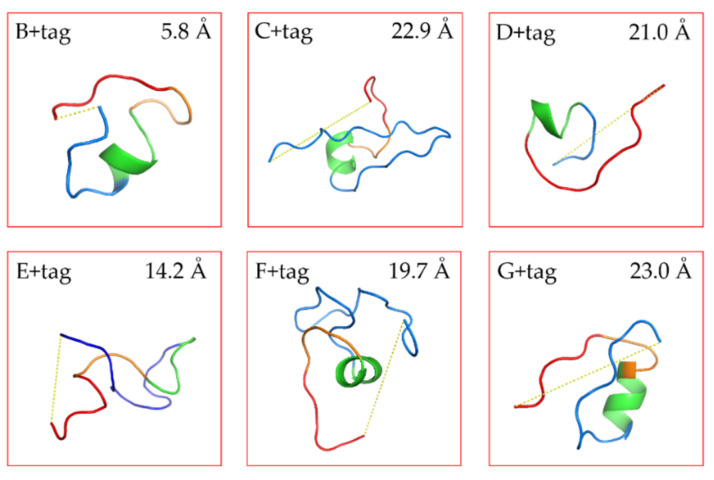
Structure prediction diagram for linkers with tag: the blue line represents the linker with its N-terminal end connected to VHH; the red line represents the C-terminal, which is the formylglycine generating enzyme (FGE) recognition sequence; the green ribbon represents the His_6_ sequence.

**Figure 2 biomolecules-10-01610-f002:**
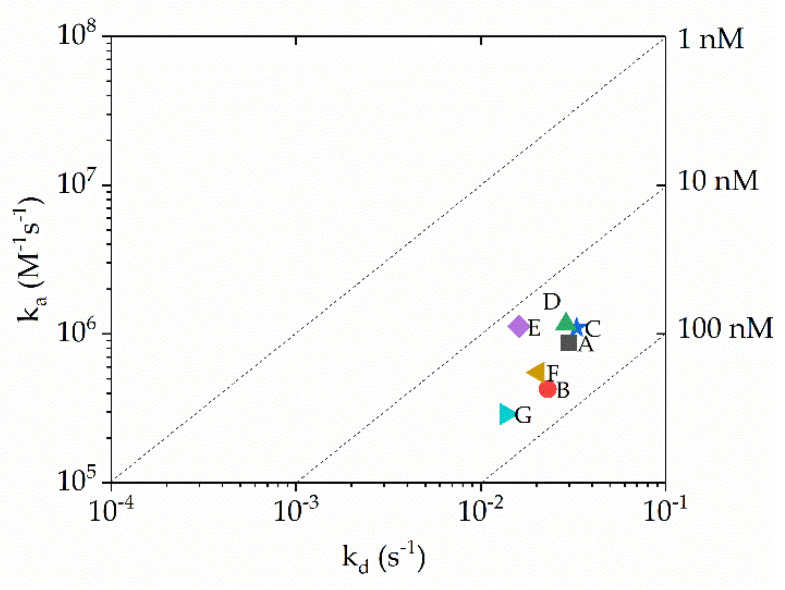
RaPID plot for the VHHs with different linkers. The kinetic rate binding constant values *ka* and *kd* are determined by surface plasmon resonance SPR and plotted on a 2D diagram, so that binders located on the same diagonal line have identical *K_D_* values.

**Figure 3 biomolecules-10-01610-f003:**
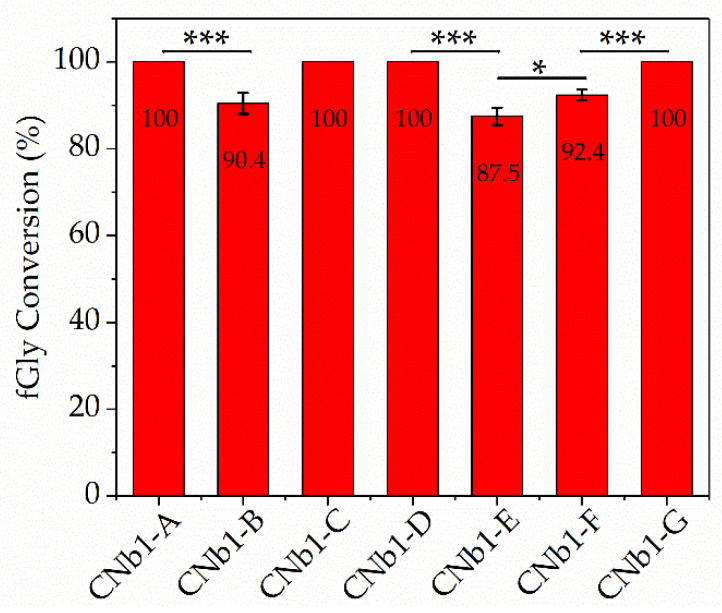
fGly conversion of the VHHs with different linkers. The VHHs were catalyzed by FGE under the same reaction conditions, then the conversion of cystine to fGly resulted in a loss of 18 Da that was identified and quantified by HPLC-HRMS. (*t*-test, * *p* < 0.05, *** *p* < 0.001).

**Figure 4 biomolecules-10-01610-f004:**
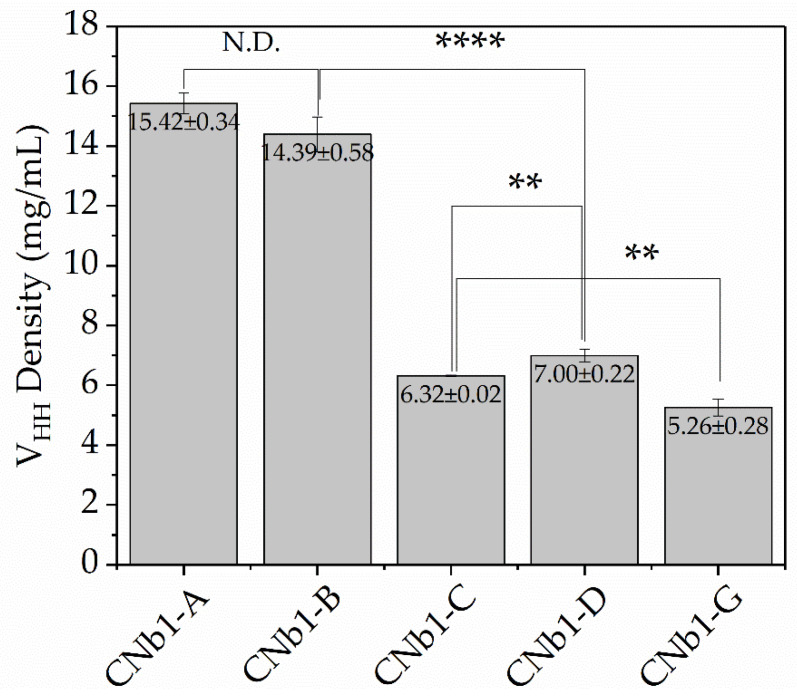
Coupling density of VHHs with different linkers. The final coupling density of VHHs upon immobilization at the same dosages of VHHs (input amount of VHHs in mol) and the same reaction conditions. (*t*-test, ** *p* < 0.01, **** *p* < 0.0001).

**Table 1 biomolecules-10-01610-t001:** Summary of yield and molecular weight of VHHs with different linkers.

VHH	Linker	Yield (mg/L)	Mass (Da)	Expression Level in Molar % Relative to No Linker (100%)
CNb1-A	-	197 ± 4	15,975.03	100
CNb1-B	AHHSEDP	191 ± 6	16,784.98	92
CNb1-C	EPKTPKPQPQPQPQPQPNPTTE	157 ± 3	18,421.83	69
CNb1-D	GGGGS	161 ± 2	16,290.61	81
CNb1-E	GGGGSGGGGSGGGGS	95 ± 5	16,956.39	45
CNb1-F	AHHSEDPGGGGSGGGGSGGGGS	208 ± 4	17,730.84	95
CNb1-G	STPPTPSPSTPP	160 ± 2	17,121.40	76

**Table 2 biomolecules-10-01610-t002:** The maximum adsorption capacity (qm) and Langmuir adsorption constant (Kd) of VHH-based immunosorbents.

VHH	qm (mg/g Resin)	Kd (M)	Activity Retention (%)
CNb1-A	5.68 ± 0.02	(2.14 ± 0.19) × 10^−5^	49.87
CNb1-B	4.83 ± 0.03	(7.51 ± 0.13) × 10^−6^	47.74
CNb1-C	2.89 ± 0.11	(9.61 ± 1.22) × 10^−6^	71.39
CNb1-D	2.49 ± 0.02	(9.18 ± 0.40) × 10^−6^	49.11
CNb1-G	2.48 ± 0.02	(7.81 ± 0.32) × 10^−6^	68.41

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
