# Peer review of "Peptide Linker Affecting the Activity Retention Rate of VHH in Immunosorbents"

_biomolecules, 2020, doi:10.3390/biom10121610_

Round 1
Reviewer 1 Report
This study describes the testing of a range of linker sequences with different properties (length/rigidity) to probe the effects of these properties on parameters such as expression yield and activity. The introduction is well written and clear. However I find the conclusions of the study inappropriate given that 2 of the linkers were not evaluated for all aspects. I am not clear why linker F was dropped. The authors state that due to yield and activity E and F were not continued with however F has yield and activity equally as good as B! In order to thoroughly evaluate different linkers as outlined in the study hypothesis all assays need to be completed on all linkers. I am struggling to see how any linker is better than no linker in the data provided.
More details could be provided in the methods including OD induced at, how cells were harvested and how protein was eluted form the his trap to aid reproducibility. Additionally the composition of the SPR buffer (HBS-EP+) needs to be provided.
The paragraph covering lines 184-189 is confusing as currently written, I am not clear what the authors are trying to say about their construct design so should be made clearer.
The authors state their proteins are made to 95% purity how is this evaluated?
How is protein yield determined? Was this determined from a single expression or repeated?
Figure 1 - It is difficult to relate discussion in the text to what can be seen in the figure e.g. models C and G do not appear more rigid or different in length as currently presented. This could be improved by providing size measurements, ensembles of models produced to show flexibility or a parameter that represents this from the package used etc.
Additional corrections:
line 145 - remove purchased from
line 146 - remove add (as already have added later in the sentence)
line 182 - remove obviously (as not needed)
line 198 - spelling error much
line 198 - authors state that constructs, C, E and G are proline and glycine rich, however all of the constructs are proline or glycine rich so this statement is not relevant
line 229 - Nanobody does not need a capital letter here
line 239 - the authors state that linkers C and G improve efficiency to 100% but no linker is already at 100% so I am not sure how this can be classed as an improvement?
line 241 - needs a inserting
line 262 - basically identical is not scientific. Figure 4 needs statistics adding onto the figure and in text discussion. This is particularly relevant as the authors talk about short length improving density but this is not evident as currently presented for linker D.
line 270 - conditiosn is incorrect
line 272 - certrainly is incorrect, this sentence is also poor grammar
Throughout the text there are areas where tenses are mixed e.g. final paragraph of results section
Start of conclusions section, the use of word conjunction is incorrect here.
The authors mention that they observe a dramatic improvement in binding in this study compared to previous reports (0.13-0.75mg) but provide no explanation as to why/how this improvement is achieved.
Reviewer 2 Report
This manuscript by Da Li, et al. reports a well-designed study on the influence of C-terminal linker sequences on VHH domain immobilization efficiency and and binding activity. They prepare recombinant VHH domains in E. coli fused C-terminally to various peptide linkers that separate the core VHH from a formylglycine tag. They tested a variety of long and short sequences with different predicted levels of flexibility/stiffness. Proline incorporation into the linker domains was postulated to increase the linker stiffness. In the end, the influence of linker addition on expression levels was slightly negative (worse expression levels for VHH domains with linkers). Additionally, the affinity was not changed and f-Gly conversion efficiency was either kept constant (compared to no linker), or slightly inhibitor. In summary, the linkers tended to make this VHH immobilization and binding capability slightly worse.
Nonetheless based on the written manuscript, this work was designed and performed carefully and the results should be valid. I have only a few minor comments / corrections.
Specific comments:
- Typo: Page 3, line 86 ‘utmaost‘ should be changed to ‘utmost‘
- Role of disulfide bond formation: VHH domains have a conserved disulfide bond. Recently work was undertaken to study the influence on VHH disulfide bond reduction / removal on binding affinity, binding mechanics, and thermal stability. (Nano Letters, DOI: 10.1021/acs.nanolett.9b02062 ) . Since they chose to express the VHH domain in the cytoplasm which tends to be reducing, the authors should comment on the presence or absence of disulfide bonds in their VHH. If different constructs have different levels of disulfide bond oxidation, it could explain some of the adsorption data. In the referenced study, removal of the disulfide bond did not decrease antigen-binding affinity, but significantly decreased the thermal stability. Since a fraction of their VHH becomes inactivated upon binding to the resin, it would be interesting to know whether different % of oxidized / reduced nanobodies could explain it. The authors can also consider adding the above reference as well.
- Table 1: the last column on the right lists ‘Expression level in % relative to no linker (100%). For linker F, the yield is reported as ‘208 mg/L’, however it is reported as 95% expression relative to no linker. Is this an error or is the relative expression level reported in molar terms? This can be made clearer.
- The authors should report the entire amino acid (or DNA) sequences for the VHH variants, including the framework and CDR2 in addition to the linker sequences and FGE tags.
Round 2
Reviewer 1 Report
I am happy that my comments have been addressed in the revised manuscript.